# Impaired Vagal Activity in Long-COVID-19 Patients

**DOI:** 10.3390/v14051035

**Published:** 2022-05-13

**Authors:** Domenico Acanfora, Maria Nolano, Chiara Acanfora, Camillo Colella, Vincenzo Provitera, Giuseppe Caporaso, Gabriele Rosario Rodolico, Alessandro Santo Bortone, Gennaro Galasso, Gerardo Casucci

**Affiliations:** 1Department of Internal Medicine, San Francesco Hospital, Viale Europa 21, 82037 Telese Terme, Italy; acanforachiara@gmail.com (C.A.); cacolella@virgilio.it (C.C.); 2Department of Neurosciences, Reproductive Sciences and Odontostomatology, University Federico II of Naples, 80131 Naples, Italy; maria.nolano@unina.it; 3Neurology Department, Skin Biopsy Laboratory, Istituti Clinici Scientifici Maugeri IRCCS, 82037 Telese Terme, Italy; vincenzo.provitera@fsm.it (V.P.); giuseppe.caporaso@icsmaugeri.it (G.C.); 4Department of Biotechnological and Applied Clinical Science, University of L’Aquila, 67100 L’Aquila, Italy; 5Department of Neurology, Careggi University Hospital, University of Florence, 50134 Florence, Italy; gabrielerodolico.work@gmail.com; 6Division of Cardiac Surgery, Department of Emergency and Organ Transplantation, University of Bari, 70124 Bari, Italy; alessandro.bortone@gmail.com; 7Department of Medicine Surgery and Dentistry, University of Salerno, 84121 Salerno, Italy; ggalasso@unisa.it

**Keywords:** Long-COVID-19, dysautonomia hypothesis, heart rate variability, procoagulative state, D-dimer, NT-ProBNP

## Abstract

Long-COVID-19 refers to the signs and symptoms that continue or develop after the “acute COVID-19” phase. These patients have an increased risk of multiorgan dysfunction, readmission, and mortality. In Long-COVID-19 patients, it is possible to detect a persistent increase in D-Dimer, NT-ProBNP, and autonomic nervous system dysfunction. To verify the dysautonomia hypothesis in Long-COVID-19 patients, we studied heart rate variability using 12-lead 24-h ECG monitoring in 30 Long-COVID-19 patients and 20 No-COVID patients. Power spectral analysis of heart rate variability was lower in Long-COVID-19 patients both for total power (7.46 ± 0.5 vs. 8.08 ± 0.6; *p* < 0.0001; Cohens-d = 1.12) and for the VLF (6.84 ± 0.8 vs. 7.66 ± 0.6; *p* < 0.0001; Cohens-d = 1.16) and HF (4.65 ± 0.9 vs. 5.33 ± 0.9; *p* = 0.015; Cohens-d = 0.76) components. The LF/HF ratio was significantly higher in Long-COVID-19 patients (1.46 ± 0.27 vs. 1.23 ± 0.13; *p* = 0.001; Cohens-d = 1.09). On multivariable analysis, Long-COVID-19 is significantly correlated with D-dimer (standardized β-coefficient = 0.259), NT-ProBNP (standardized β-coefficient = 0.281), HF component of spectral analysis (standardized β-coefficient = 0.696), and LF/HF ratio (standardized β-coefficient = 0.820). Dysautonomia may explain the persistent symptoms in Long COVID-19 patients. The persistence of a procoagulative state and an elevated myocardial strain could explain vagal impairment in these patients. In Long-COVID-19 patients, impaired vagal activity, persistent increases of NT-ProBNP, and a prothrombotic state require careful monitoring and appropriate intervention.

## 1. Introduction

Patients discharged from hospital after acute COVID-19 have an increased risk of multiorgan dysfunction, readmission, and mortality [1]. The term “Long-COVID-19” refers to the signs and symptoms that continue or develop after the “acute COVID-19” phase and include both “ongoing symptomatic COVID-19” and “post COVID-19 syndrome” [2]. Recent joint guidelines proposed by the Scottish Intercollegiate Guidelines Network (SIGN), the National Institute for Health and Care Excellence (NICE), and the Royal College of General Practitioners (RCGP) have divided COVID-19 infection into 3 phases—“Acute COVID-19” (signs and symptoms of COVID-19 infection up to 4 weeks), “ongoing symptomatic COVID-19” (from 4 weeks up to 12 weeks), and “post-COVID-19 syndrome” (when signs and symptoms continue beyond 12 weeks) [2]. In addition to the persistence of symptoms, it is also possible to detect abnormalities of chest radiographs and biomarkers [3]. Interestingly, in Long-COVID-19 patients, it is possible to detect a prolonged elevation of D-dimer, regardless of the inflammatory indices and the severity of the acute phase [4]. In these patients with persistent elevation of the D-dimer, there is an increase in serious thromboembolic complications [4,5]. It has been recently reported that many symptoms in Long-COVID-19 patients may be due to autonomic nervous system impairment [6]. Heart rate variability (HRV) analysis is among the non-invasive methods that can be used to assess cardiac autonomic activity [7]. HRV can be calculated using 24-h ECG monitoring recordings [8]. HRV includes time-domain measures (standard deviation of the normal-to-normal (NN) interval (SDNN), standard deviation of the average NN interval (SDANN) calculated over 5-min periods, average standard deviation of the averages of all normal-to-normal R-R intervals in all 5-min segments of the entire recording (ASDNN), square root of the mean squared differences of successive NN intervals (rMSSD), and ratio between the number of interval differences of successive NN intervals of more than 50 ms and the total number of NN intervals (pNN50) and frequency-domain measures (very low frequency (VLF) at a frequency between 0.0033 and 0.04 Hz, low frequency (LF) at a frequency between 0.04 and 0.15 Hz, high frequency (HF) at a frequency between 0.15 and 0.4 Hz, and low-frequency/high-frequency ratio (LF/HF). All HRV parameters are impacted by the autonomic nervous system and dominantly controlled by the parasympathetic nervous system (PNS) [6,9,10]. SDNN has been found to be correlated with VLF and LF [10]. SDNN is also suggested as the gold standard for medical stratification of cardiac risk in 24-h ECG recordings since it provides information about cardiac morbidity and mortality [7]. PNN50 was found to be correlated with RMSSD and HF [8]. VLF is associated with arrhythmia-related deaths [9]. VLF was also found to be more closely related to all-cause mortality in comparison with LF and HF [9,10]. All these parameters are easy, non-invasive, inexpensive, and repeatable.

The sympathetic nervous system promotes pro-inflammatory responses through the release of catecholamines and beta-adrenergic stimulation, while the parasympathetic nervous system promotes anti-inflammatory effects [11]. COVID-19 and Long-COVID-19 can be systemic diseases associated with systemic inflammation and procoagulative state, and can promote sympatho/vagal imbalance, revealed by acute and convalescent signs and symptoms of the disease.

The aim of this study is to verify the dysautonomia hypothesis in Long-COVID-19 patients and to evaluate the relationship between heart rate variability, inflammation, and procoagulative state.

## 2. Materials and Methods

During a study on the characterization of the clinical picture, laboratory findings, and prognosis of patients with Long-COVID-19 [12], an analysis of HRV was performed to assess the relationship between autonomic dysfunction and Long-COVID-19 in patients who developed the disease [5]. In this study, we analysed HRV data to investigate the correlation between thrombosis, inflammation, and HRV. We consecutively enrolled 50 patients admitted to our Department from 1 May 2021 to 30 June 2021 with symptoms including dyspnea, fatigue, cough, headache, loss of appetite, and myalgia. Exclusion criteria were the following: permanent atrial fibrillation, ventricular ectopic beats >600/24 h, presence of a pacemaker or implantable cardioverter defibrillator, and refusal to participate in the study. This study was performed in accordance with the ethical guidelines of the 1975 Declaration of Helsinki and was approved by the local Institutional Review Board (22 March 2021) of the San Francesco Hospital of Telese Terme (BN). All procedures complied with the ethical standards of institutional and national research. Signed informed consent was obtained from all participants included in the study. Of the patients enrolled, thirty recovered from COVID-19.

### 2.1. Study Population

All patients were clinically stable, physically inactive, and in sinus rhythm. Patients had been on a stable pharmacological regimen for at least 2 weeks before hospital admission. Therefore, all patients were on optimal medical therapy at the time of admission. After admission, all patients underwent demographics, medical history, vital signs, clinical and anthropometric evaluations, blood chemistry tests, and standard echocardiography. In addition, the Severity of COVID-19 WHO Clinical Classification [13] scale was administered to patients in the acute phase of COVID-19, and the Post-COVID-19 Functional Scale [14] was administered to patients who recovered from acute COVID-19.

### 2.2. Heart Rate Variability Assessment

A 24-h HRV analysis was previously validated in our laboratory [15]. Patients underwent a 24-h ECG recording by a portable twelve-channel tape recorder, processed by a DM Software Inc. version 12.5.0078a, (Diagnostic Monitoring Software P.O. Box 3109; 209 Kingsbury Grade, #3; Stateline, NV 89449, USA). The 24-h ECG monitoring recordings were exported to the PC, manually scanned to remove any artifacts, and analyzed with the 24-h ECG software with a sampling frequency of 128 Hz. All recordings were performed at admission; after preparation of the skin, self-adhesive electrodes were placed in the positions usually used for twelve-lead ECG monitoring, and recording was started between 9:00 and 9:30 a.m. During the recording period, patients were allowed to be standing or sitting next to their beds, while other activities were not allowed. HRV parameters were automatically documented by the 24-h ECG monitoring software as numerical data. In order to be considered eligible for the study, each recording had to have at least 12 h of analyzable intervals between consecutive R peaks. Moreover, the analyzable recording period had to include at least half of the night-time (from 00:00 a.m. through to 5:00 a.m.) and half of the daytime (from 7:30 a.m. through to 11:30 p.m.) [7]. Each beat was labelled as normal or aberrant according to recognition by the algorithm for tape analysis and after an investigator’s verification. Time-domain measures (SDNN, SDANN, ASDNN, rMSSD, pNN50) and frequency-domain measures (VLF, LF, HF, LF/HF) were calculated for HRV assessment. Power spectral density (PSD), a highly reproducible tool to assess the functional balance between parasympathetic and sympathetic domains of the autonomic nervous system activity and to decompose the total variation of a data series into its frequency components, was estimated using the Blackman–Tukey method in all accepted segments after linear trend removal. The total power (TP) and the power in the very low frequency band (VLF, 0.01–0.04 Hz), low frequency band (LF, 0.04–0.15 Hz), and high frequency band (HF, 0.15–0.45 Hz) were then computed by numerical integration of the spectral density function. Only normalized TP, VLF, LF, and HF values were considered in the analysis and were expressed as natural logarithms (ln ms^2^) to minimize skewness. The LF/HF ratio was also calculated. The following HRV parameters were defined in accordance with the ACC/AHA/ESC consensus [8]: HF component, which mainly reflects the efferent vagal activity; LF component, mediated by efferent vagal and sympathetic activity; LF/HF ratio, a measure of sympatho/vagal balance; VLF component, reflecting neuroendocrine and thermoregulatory influences mediated mainly by the sympathetic system [9,10]. The most commonly used time-domain parameters were derived from normal RR intervals (NN). Among these, the square root of the mean squared differences of successive RR intervals (RMSSD) is associated with respiratory effects on heart rate and is modulated by both parasympathetic and sympathetic activity, with the proportion derived by dividing the number of interval differences of successive NN intervals greater than 50 ms (NN50) by the total number of NN intervals (pNN50) reflects rapid adjustments, and the standard deviation of NN (SDNN) expresses overall HRV regulation [8].

### 2.3. Statistical Analysis

Normal distribution of continuous variables was tested by the Kolmogorov–Smirnov test. Continuous variables were expressed as means ± standard deviations (SD). Differences between groups were analyzed for statistical significance with the unpaired *t*-test, Mann–Whitney *U* test, or χ^2^ analysis, as appropriate. Variables that were not normally distributed were transformed to normalize them, if feasible. The absolute powers and the spectral components were presented as natural logarithms (ln ms^2^) to minimize skewness. Cohens-d effect size [16] was calculated for differences in continuous variables between patients with and without Long-COVID-19.

Multivariable linear regression analyses were performed to examine the independent correlates between Long-COVID-19 status and parameters of thrombosis, inflammation, NT-ProBNP, and HRV assessment, and was adjusted for demographics, medical history, pre-existing conditions, vital signs, therapies, and echocardiography measurements. *p*-value < 0.05 was considered as statistically significant. All statistical analyses were carried out using R (2013) (http://www.R-project.org) (accessed on 30 September 2021) [17].

## 3. Results

Fifty consecutive patients were enrolled in this study, of which 30 were Long-COVID-19 and 20 were No-COVID-19. Other demographic data of the population analyzed in this study were previously reported [5]. In the 30 Long-COVID-19 patients, the mean duration of nasopharyngeal swab positivity for SARS-CoV-2 was 23.1 ± 8 (range 11–49) days. During the acute phase of COVID-19, according to the Severity of COVID-19 WHO Clinical Classification [13], 21 patients were classified as Mild/Moderate and 9 as Severe/Critical. On the Post-COVID-19 Functional Status Scale [14], 7 (23.3%) patients reported No/Negligible functional limitations, 6 (20%) Slight Functional Limitations, and 17 (56.7%) Moderate/Severe Functional Limitations. The patients included in this HRV study had comparable socio-demographic characteristics, but echocardiographic findings showed that left ventricular ejection fraction was lower in Long-COVID-19 patients (Table 1).

The inflammatory parameters and coagulation pathway were higher in Long-COVID-19 patients (Table 2).

A good quality recording of HRV parameters was available for all patients enrolled in this study. Table 3 summarizes the 24-h ECG monitoring parameters in Long-COVID-19 and No-COVID-19 patients.

The minimum heart rate was lower in No-COVID-19 patients (*p* = 0.031; Cohens-d = 0.64). Time-domain heart rate variability shows lower SDNN in Long-COVID-19 patients (*p* = 0.0001; Cohens-d = 1.12) and consequently lower SDANN (*p* = 0.001; Cohens-d = 1.02) and SDNNi values (*p* = 0.001; Cohens-d = 1.05) (Table 3). Power spectral analysis of the heart rate variability was lower in Long-COVID-19 patients both for total power (*p* = 0.0001; Cohens-d = 1.12) and for the VLF (*p* = 0.0001; Cohens-d = 1.16) and HF components (*p* = 0.015; Cohens-d = 0.76). LF/HF ratio was significantly higher in Long-COVID-19 patients (*p* = 0.001; Cohens-d = 1.09). By stratifying patients in Long-COVID-19 and No-COVID-19, SDNN was lower in Long-COVID-19, while LF/HF ratio, D-dimer, NT-Pro-BNP, and IL-6 were significantly higher in Long-COVID-19 patients (Figure 1).

On multivariable analysis, Long-COVID-19 is independently and significantly correlated with D-dimer (standardized β-coefficient = 0.259), NT-ProBNP (standardized β-coefficient = 0.281), HF component of spectral analysis (standardized β-coefficient = 0.696), and LF/HF ratio (standardized β-coefficient = 0.820) (Table 4).

## 4. Discussion

Our study shows that HRV was significantly different in Long COVID-19 compared to No-COVID-19 patients. Our population did not show any significant differences between Long-COVID-19 and No-COVID-19 in demographics, medical history, drug use, and vital signs. In this study, the enrolled population showed no gender differences, while previous studies reported that females are most affected by Long-COVID-19 [18]. Further studies are, therefore, needed to verify gender differences in autonomic dysfunction in Long-COVID-19 patients.

In this study, the HRV, in time and frequency domains, a non-invasively measure of sympathetic and parasympathetic activities, was used to highlight the dysautonomia hypothesis in Long COVID-19 patients. HRV is one of the main markers of dysautonomia [19]. HRV could be an important tool to better understand the inflammatory mechanism and neuroimmune system involved in Long-COVID-19 patients. We found a significant correlation between Long-COVID-19 syndrome and HF component of spectral analysis and LF/HF ratio, suggestive of vagal impairment. Some pathophysiological mechanisms may explain dysautonomia in Long-COVID-19 patients including neurotropism, procoagulative state, and inflammation. However, it remains unclear whether dysautonomia associated with Long COVID-19 directly results from post-infectious immune-mediated processes or from the autonomic-virus pathway. In the acute phase, it has been shown that SARS-CoV-2 virus exploits the angiotensin-converting enzyme 2 (ACE2) receptor to enter the cells [20] (Figure 2).

ACE2 is predominantly expressed in human airway epithelia, lung parenchyma, vascular endothelia, kidney cells, and small intestine cells [23,24,25]. However, ACE2 receptors are also expressed in both neurons and glia, particularly in the brainstem and in the regions responsible for cardiovascular function regulation (subfornical organ, paraventricular nucleus, nucleus of the tractus solitarius, and rostral ventrolateral medulla) [23,25]. SARS-CoV-2 neuronal invasion leads to stimulation of the adrenal glands and release of epinephrine and norepinephrine, the so called “catecholamine surge,” which results in cardiac injury and altered contractility and constriction on vessels, leading to increased systemic vascular resistance. This leads to increased fluid pressure and alveolar leakage and increased blood pressure in pulmonary arteries and capillaries, resulting in capillary leakage into the alveoli. Additionally, brain injury can lead to hyperventilation and endotracheal intubation with mechanical ventilation, thereby causing acute lung injury and acute respiratory distress syndrome. Finally, viral brain invasion causes a blood-brain barrier breakdown that results in the release of immune cells, cytokines, and damage-associated molecular patterns, which may lead to lung endothelial and epithelial damage that initiates or exacerbates lung injury. The finding of neurofilaments light chain protein (NfL), an expression of axonal damage, and of glial fibrillary acidic protein (GFAp), an expression of astrocytic activation/injury, in the plasma of COVID-19 patients demonstrates the damage induced by the virus on the nervous system [26,27] (Figure 3).

GFAp is also detectable in patients with moderate disease [26]. In patients with severe COVID-19 disease, GFAp values are significantly reduced during the course of the disease, while NfL levels increase [27]. This evidence shows that SARS-CoV-2 has a long persistence in neurons. Detection of SARS-CoV-2 in the vagus nerve supports viral trafficking between the brainstem, lung, heart, and other organs innervated by the vagus (Figure 1) [29]. Parasympathetic impairment could be secondary to persistent vagal fibre damage in Long-COVID-19 patients.

We found a significant association between Long-COVID-19 and increased plasma concentrations of NT-ProBNP. Plasma levels of NT-proBNP reflect myocardial strain due to increased pressure [30,31]. The persistence of elevated levels of NT-ProBNP, an expression of myocardial strain, may contribute to dysautonomia in Long-COVID-19 patients. However, levels may also increase in response to other insults such as ischemia or inflammatory cytokines. NT-ProBNP was shown to predict adverse outcomes in patients with COVID-19 [30].

Our results demonstrate that Long-COVID-19 patients had higher expression of cardiac injury biomarkers and more severe coagulation dysfunction than No-COVID-19 patients. In Long-COVID-19 patients, it is possible to detect a prolonged elevation of D-dimer, regardless of the inflammatory indices and the severity of the acute phase. In these patients with persistent elevation of the D-dimer, there is an increase in serious thromboembolic complications [4]. Long-COVID-19 patients had no evidence of hypofibrinogenemia or thrombocytopenia. These results suggest that neither low-grade disseminated intravascular coagulopathy nor systemic coagulation activation can explain the elevation in D-dimer levels in our Long-COVID-19 patients.

Some studies have shown that COVID-19 contributes to arrhythmias [32,33], which may eventually be due to impaired autonomic nerve function.

## 5. Limitations

Due to the relatively small sample size, which limits the statistical power of these analyses, we cannot exclude the possibility that some findings results were non-significant due to lack of statistical power. The main strength of our study was the use of a validated parameter to estimate HRV. Standardized protocols also make our study very valuable. Few studies have investigated dysautonomia in Long-COVID-19 patients using different methods [6,34]; therefore, a comparison with previous results is difficult.

## 6. Conclusions

To our knowledge, this study is the first to analyze HRV in Long-COVID-19 patients using power spectral analysis. Dysautonomia may explain the persistent symptoms in Long-COVID-19 patients. The persistence of a procoagulative state and an elevated myocardial strain could explain vagal impairment in these patients. An evaluation of cholinergic nerve fiber damage in Long-COVID-19 patients at present could ultimately confirm the impairment of vagal activity. A direct in vivo evaluation of autonomic innervation in Long-COVID-19 patients is ongoing in our laboratories. Our findings may have an important clinical implication. An adequate therapeutic approach [5,35] could improve the clinical picture and prognosis of Long-COVID-19 patients. In Long-COVID-19 patients, impaired vagal activity, persistent increases of NT-ProBNP, and a prothrombotic state require careful monitoring and appropriate intervention.

## Figures and Tables

**Figure 1 viruses-14-01035-f001:**
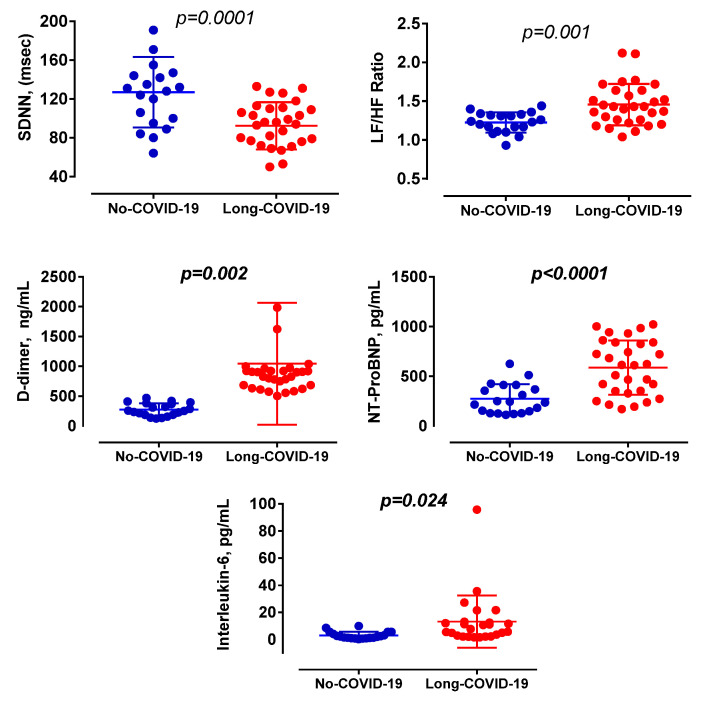
Main relationship of HRV (SDNN, LF/HF ratio), D-Dimer, NT-ProBNP, and IL-6 in No-COVID-19 compared to Long-COVID-19 patients.

**Figure 2 viruses-14-01035-f002:**
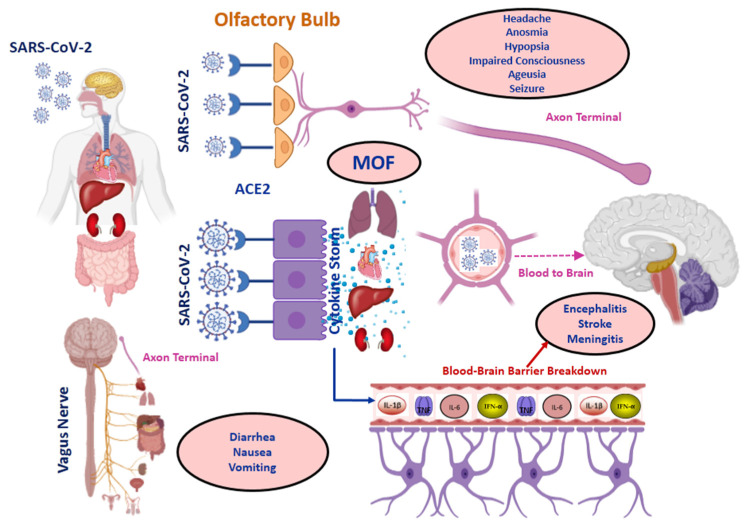
Routes of SARS-CoV-2 invasion. SARS-CoV-2 is mainly transmitted from one person to another by inhalation of droplets. SARS-CoV-2 enters the mucosal cells of the respiratory tract, conjunctiva, and gastrointestinal tract through ACE2 receptors. When the virus comes into contact with the ocular conjunctiva, it could reach the central nervous system via the trigeminal nerve. Although the hypothesis is still controversial, some authors believe that when SARS-CoV-2 comes into contact with the nasal mucosa, it reaches the brain through the olfactory nerve and that the vagus nerve, which innervates the respiratory system, the heart, the digestive system, the kidneys, bladder, uterus, and testicles, is a large route of transfer to the central nervous system. The virus enters the brain via neuronal retrograde transport up to the axonal terminal. SARS-CoV-2 also invades COVID-19 patients through the vasculature and lymphoid pathways. Once the virus has entered the circulation, it can invade the brain through blood-brain barrier breakdown. When the virus comes into contact with the host cell, the innate immune response activates the cytokine storm, particularly during ARDS hypoxia in patients with severe COVID-19. Cytokine storm leads to multi-organ failure (MOF) and damaged blood-brain barrier. With an intact blood-brain barrier, the passage of SARS-CoV-2 to the brain is unlikely [21,22]. This figure was created using the website https://app.biorender.com (accessed on 14 October 2021).

**Figure 3 viruses-14-01035-f003:**
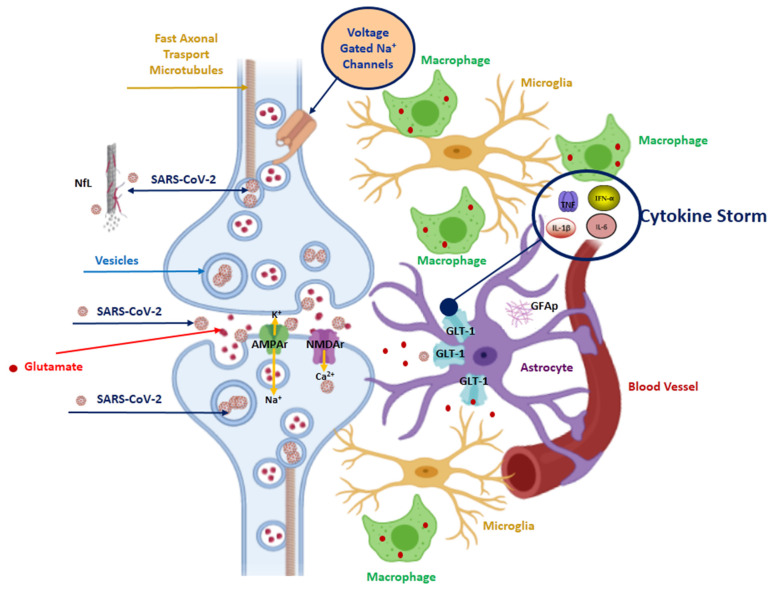
SARS-CoV-2 spreads by transsynaptic viral neuroinvasion from periphery to the brain. The retrograde transsynaptic viral spread occurs via a mechanism of endocytosis or exocytosis and the transport of vesicles occurs along the fast axonal microtubules. Axonal damage is expressed by the increased concentration of NfL in COVID-19 patients. The neuroinvasion of SARS-CoV-2 induces an excess of glutamate at the synaptic level. Moreover, high levels of inflammatory cytokines such as tumour necrosis factor (TNF) and interleukin (IL)-1β released by activated inflammatory cells, including microglia, astroglia, and macrophages, lead to increases in synaptic glutamate concentrations. SARS-CoV-2 damages macrophages, microglia, and astrocytes. In fact, in COVID-19 patients even with moderate disease, it is possible to find increased glial fibrillary acid protein (GFAp) as a marker of astrocytic activation/injury. Effects of inflammatory molecules on astrocytic cell morphology leads to decreased ability to sequester and contain glutamate within the synapse, resulting in a spill-over of the glutamate into the extrasynaptic space. Increases in synaptic glutamate resulting from early inflammatory changes induced by SARS-CoV-2 invasion have been shown to induce overactivation of intrasynaptic ionotropic receptors, such as α-amino-3-hydroxy-5-methyl-4-isoxazolepropionic acid (AMPA) and N-methyl-d-aspartate receptors (NMDA), potentially contributing to excitotoxicity [28]. This figure was created using the website https://app.biorender.com (accessed on 14 October 2021).

**Table 1 viruses-14-01035-t001:** Clinical characteristics and baseline values of the study population.

Demographics, Medical History and Vital Signs	Long-COVID-19	No-COVID-19	Effect Size Cohens-d ^b^
Number of patients, *n*	30	20	
Sex, M/F, *n*	17/13	8/12	
Age, years ^a^	58.6 ± 17.6	56.3 ± 14.7	0.14
Weight, kg ^a^	77.1 ± 14.5	73.8 ± 12	0.25
Height, cm ^a^	164.6 ± 11.4	169.1 ± 8.7	0.44
Body mass index, kg/m^2 a^	28.4 ± 4.2	25.7 ± 2.4	0.79
Pre-existing conditions in the last year, n (%)			
Cancer	2 (6.7%)	1 (5.0%)	
Chronic heart disease	13 (43.3%)	6 (30.0%)	
Chronic kidney disease	5 (16.6%)	2 (10.0%)	
Chronic liver disease	3 (10.0%)	1 (5.0%)	
Chronic lung disease	7 (23.3%)	7 (35.0%)	
Chronic neurological disease	9 (30.0%)	5 (25.0%)	
Diabetes	7 (23.7%)	3 (15.0%)	
Hypertension	19 (63.3%)	11 (55.0%)	
Mental health conditions	2 (6.66%)	1 (5.0%)	
Obesity (Body Mass Index > 30)	11 (36.6%)	3 (15.0%)	
Heart rate, bpm ^a^	73 ± 15	70 ± 13	0.21
Systolic blood pressure, mmHg ^a^	121 ± 15	121 ± 17	0
Diastolic blood pressure, mmHg ^a^	78 ± 12	76 ± 10	0.18
Therapies, n (%)			
ACE-I/ARB/ARNIs	19 (63%)	12 (60%)	
Beta-blockers	11 (37%)	8 (40%)	
ASA	13 (43%)	9 (45%)	
Diuretics	11 (37%)	6 (30%)	
Anticoagulants	12 (40%)	6 (30%)	
Echocardiography Measurements			
LV end diastolic dimension, cm ^a^	4.8 ± 1	4.5 ± 0.6	0.36
LV end diastolic volume, mL ^a^	114.6 ± 52.5	94.1 ± 27.9	0.49
LV end systolic dimension, cm ^a^	3.2 ± 1.04	2.6 ± 0.5 *	0.73
LV end systolic volume, mL ^a^	48.7 ± 38.5	28 ± 10.5 ^†^	0.73
LV ejection fraction, % ^a^	61.9 ± 13.7	70.4 ± 5.7 •	0.81
Left atrial anteroposterior dimension, cm ^a^	3.7 ± 1.3	3.5 ± 0.5	0.20
E/A ratio ^a^	1.02 ± 0.4	1.1 ± 0.3	0.22
SPAP, mmHg ^a^	13.8 ± 10.5	14.6 ± 8.6	0.08

M = Male; F = Female; bpm = beats per minute; ACE-I = angiotensin-converting enzyme inhibitor; ARB = angiotensin receptor blocker; ARNIs = Angiotensin Receptor Neprilysin Inhibitors; ASA = Acetylsalicylic Acid; LV = Left Ventricular; PAP = Systolic Pulmonary Artery Pressure. ^a^ Mean ± standard deviation. ^b^ Cohens-d: small (0.2–0.5), moderate (0.5–0.8), and large effect size (>0.8). * refers to *p* = 0.023; ^†^ refers to *p* = 0.024; • refers to *p* = 0.012.

**Table 2 viruses-14-01035-t002:** Laboratory data of the study population.

Laboratory Values (Reference Range)	Long-COVID-19	No-COVID-19	Effect Size Cohens-d ^b^
White Blood Cell count (3.7–10.3), ×10^9^/L ^a^	6.84 ± 2.6	7.14 ± 2.3	0.12
Red Blood Cell count (4.0–10.0), ×10^6^/L ^a^	4.53 ± 0.6	4.8 ± 0.58	0.46
Haemoglobin (13.7–17.5), g/dL ^a^	14.9 ± 6.4	14.2 ± 1.8	0.15
Platelet count (155–369), ×10^9^/L ^a^	221 ± 92	244 ± 50	0.31
Prothrombin time (9.6–12.5), second ^a^	14.2 ± 2.5	13.5 ± 1.2	0.36
International normalized ratio (0.9–1.2) ^a^	1.07 ± 0.2	1.00 ± 0.09	0.45
Activated Partial Thromboplastin Time (19–30), s ^a^	30.6 ± 5.1	28.8 ± 2.6	0.44
Fibrinogen (150–450), mg/dL ^a^	364.8 ± 154.4	326.9 ± 86.1	0.30
Lactate dehydrogenase (140–280), U/L ^a^	448.1 ± 133	342.45 ± 90.5 *	0.93
Creatinine (0.8–1.30), mg/dL ^a^	0.92 ± 0.25	0.86 ± 0.23	0.25
Aspartate Aminotransferase (0–31), U/L ^a^	25.04 ± 12.2	21.6 ± 12.2	0.28
Alanine Aminotransferase (0–34), U/L ^a^	25.2 ± 14.5	20.9 ± 14.6	0.3
High Sensitivity C Reactive Protein (0–45), mg/L ^a^	16.3 ± 50.1	3.95 ± 8.8	0.34
Sodium (135–155), mEq/L ^a^	139 ± 2.7	139 ± 2.02	0
Potassium (3.5–5.5), mEq/L ^a^	4.1 ± 0.27	4.3 ± 0.4	0.59
D-dimer (250–500), ng/mL ^a^	1044.4 ± 1022	273.7 ± 106 ^†^	1.06
Erythrocyte Sedimentation Rate (0–15), mm ^a^	25.7 ± 33.2	15.5 ± 17.2	0.38
Albuminuria (0–2.5), mg/dL ^a^	120.7 ± 134.7	64.6 ± 17.7	0.58
Interleukin-6 (0–6.4), pg/mL ^a^	13.2 ± 3	3 ± 2.7 •	3.58
High-sensitivity Cardiac Troponin (<19), ng/mL ^a^	9 ± 26.3	1.6 ± 0.3	0.4
NT-ProBNP (<450), pg/mL ^a^	587.4 ± 273	273.5 ± 147.9 ^◊^	1.43
SARS-CoV-2 Anti-Spike IgM (<1), EU/mL ^a^	12.2 ± 35.5	1.04 ± 2.4	0.44
SARS-CoV-2 Anti-Spike IgG (<10), EU/mL ^a^	91.5 ± 130.1	35.9 ± 61.5	0.54
Serum Ferritin (20–300), ng/mL ^a^	144.6 ± 158.6	113 ± 85.7	0.3

^a^ Mean ± standard deviation; ^b^ Cohens-d: small (0.2–0.5), moderate (0.5–0.8), and large effect size (>0.8); * refers to *p* = 0.004; ^†^ refers to *p* = 0.002; • refers to *p* = 0.024; ^◊^ refers to *p* < 0.0001.

**Table 3 viruses-14-01035-t003:** Twenty-four-hour ECG monitoring in Long-COVID-19 and No-COVID-19 patients.

	Long-COVID-19	No-COVID-19	Effect Size Cohens-d ^b^
Average Heart Rate (beats/min) ^a^	72.6 ± 12.4	67.1 ± 7.2	0.54
Minimum Heart Rate (beats/min) ^a^	53.4 ± 8.0	48.5 ± 7.4 *	0.64
Maximum Heart Rate (beats/min) ^a^	112.9 ± 20.8	108.5 ± 23.7	0.20
Supraventricular ectopic beats (ln + 1) ^a^	4.6 ± 2.3	4.16 ± 2.2	0.19
Ventricular Ectopic Beats (ln + 1) ^a^	4.6 ± 2.6	3.0 ± 2.0	0.69
Maximum QT (msec) ^a^	464.97 ± 44.5	462 ± 83.2	0.04
Maximum QTc (msec) ^a^	488.5 ± 38.2	488.6 ± 79.7	0.01
Heart Rate Variability (Time Domain)			
SDNN (msec) ^a^	92.3 ± 24.4	127 ± 36.4 ^†^	1.12
SDANN (msec) ^a^	79 ± 21.9	109.9 ± 36.8 ^•^	1.02
SDNNi ^a^	41.9 ± 15.3	57.6 ± 14.5 ^•^	1.05
rMSSD (msec) ^a^	24.5 ± 12.3	33.9 ± 20.9	0.55
pNN50 (%)^a^	5.7 ± 7.8	10.8 ± 11.2	0.53
Heart Rate Variability (Spectral Power)			
Total Power (ln msec^2^) ^a^	7.46 ± 0.5	8.08 ± 0.6 ^◊^	1.12
VLF (ln msec^2^) ^a^	6.84 ± 0.8	7.66 ± 0.6 ^◊^	1.16
LF (ln msec^2^) ^a^	6.55 ± 0.42	6.44 ± 0.74	0.18
HF (ln msec^2^) ^a^	4.65 ± 0.9	5.33 ± 0.9 ^••^	0.76
LF/HF Ratio ^a^	1.46 ± 0.27	1.23 ± 0.13 ^•^	1.09

^a^ Mean ± standard deviation; ^b^ Cohens-d: small (0.2–0.5), moderate (0.5–0.8), and large effect size (>0.8); ln = logarithm; SDNN = standard deviation of all R-R intervals; SDANN = standard deviation of the averages of R-R intervals in all 5 min segments of the entire recording; SDNNi = mean of the standard deviations of all R-R intervals for all 5 min segments of the entire recording; rMSSD = square root of the mean of the sum of the squares of differences between adjacent R-R intervals; pNN50 = percentage of difference between adjacent normal R-R intervals that is greater than 50 ms computed over the entire 24-h ECG recording; VLF = very-low-frequency component; LF = low-frequency component; HF = high-frequency component. * refers to *p* = 0.031; ^†^ refers to *p* = 0.0001; • refers to *p* = 0.001; ^◊^ refers to *p* < 0.0001; ^••^ refers to *p* = 0.015.

**Table 4 viruses-14-01035-t004:** Multivariable analysis for Long-COVID-19.

	Standardized β-Coefficient	*p*
D-dimer (250–500), ng/mL	0.259	0.047
NT-ProBNP (<450), pg/mL	0.281	0.043
HF (ln msec^2^)	0.696	0.029
LF/HF Ratio	0.820	0.002

ln = logarithm; HF = high-frequency component.

## Data Availability

The dataset generated and analysed during the current study is available from Dr. Domenico Acanfora, Department of Internal Medicine San Francesco Hospital, Viale Europa 21, 82037 Telese Terme, Benevento, Italy, E-mail: domenico.acanfora29@gmail.com.

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
