# Peer review of "Impaired Vagal Activity in Long-COVID-19 Patients"

_viruses, 2022, doi:10.3390/v14051035_

Round 1

Reviewer 1 Report

The manuscript improved significantly. However, my comment Women are more affected by long-COVID than men. Any sex differences in your study? If not, I suggest a brief discussion about sex differences (e.g. future studies) was not addressed sufficiently.

Author Response

We thank this Reviewer for the constructive comments and suggestions. Furthermore, we would really like to thank him/her for his/her appreciation about our research in the introduction section of his/her comments. Please find our point-to-point reply below.

The manuscript improved significantly. However, my comment Women are more affected by long-COVID than men. Any sex differences in your study? If not, I suggest a brief discussion about sex differences (e.g. future studies) was not addressed sufficiently.

Done.

We have no sex differences in our study population. On line 232 we added "In this study, the enrolled population showed no gender differences, while previous studies reported that females are most affected by Long-COVID-19 [18]. Further studies are, therefore, needed to verify gender differences in autonomic dysfunction in Long-COVID-19 patients." and reference 18 Pelà G, Goldoni M, Solinas E, Cavalli C, Tagliaferri S, Ranzieri S, Frizzelli A, Marchi L, Mori PA, Majori M, Aiello M, Cor-radi M, Chetta A. Sex-Related Differences in Long-COVID-19 Syndrome. J Womens Health (Larchmt). 2022 Mar 25. doi: 10.1089 / jwh.2021.0411. Epub ahead of print. PMID: 35333613.

Reviewer 2 Report

I think the article can be improved with a more focused introduction and methods. Currently, these portions are not easily understandable. The clinical significance of HRV is not clear and methods to perform are difficult. Authors' recommendations regarding performing HRV are not clear.

Author Response

We thank this Reviewer for her/his appreciating our paper. We thank this Reviewer for the constructive comments and suggestions. Furthermore, we would really like to thank him/her for his/her appreciation about our research in his/her comments. Please find our point-to-point reply below. I think the article can be improved with a more focused introduction and methods. Currently, these portions are not easily understandable. The clinical significance of HRV is not clear and methods to perform are difficult. Authors' recommendations regarding performing HRV are not clear. DoneWe have reviewed the introduction and methods and added to line 113 "A 24-hour HRV analysis was previously validated in our laboratory [15]” and reported the reference Antonelli Incalzi R, Corsonello A, Trojano L, Pedone C, Acanfora D, Spada A, D'Addio G, Maestri R, Rengo F, Rengo G. Heart rate variability and drawing impairment in hypoxemic COPD. Brain Cogn. 2009 Jun; 70 (1): 163-70. Doi: 10.1016 / j.bandc. 2009.01.010 Epub 2009 Mar 3. PMID: 19261365 to make the HRV methods more understandable.

This manuscript is a resubmission of an earlier submission. The following is a list of the peer review reports and author responses from that submission.

Round 1

Reviewer 1 Report

The manuscript under review looks at heart rate variability and laboratory parameters in 30 long-COVID-19 patients and 20 controls. The authors find lower heart rate variability and higher D-dimer and NT-ProBNP in long-COVID-19 when compared to controls.

Overall, the experiments are well designed, the methods are described thoroughly.

While this is certainly an interesting study, I have some concerns on the composition of the control cohort and severe concerns regadring the presentation of the data, especially of data that is not part of this study. Also the interpretation of the results needs to be improved.

  1. Composition of the cohorts: The controls are more or less matched regarding basic demographic data yet they have much less chronic heart disease and hypertension when compared to long-COVID-19 patients. This is likely to influence heart rate variability.
  2. Presentation of the data: The results regarding heart rate variability and laboratory parameters are not really described or put into context. Why were these parameters assessed? What is the connection between heart rate variability and D-dimer or NT-ProBNP?
  3. Presentation of data that are not part of the study: Figure 2 and 3 describe data which are not part of this study and the connection of these data to the data of the study are not obvious, additionally some statements cannot be made.

In Figure 2, routes of neuroinvasion are described as if this part of COVID-19 is fully solved. It is not and some of the staments are wrong or not backed by data. The authors write: “When the virus comes into contact with the ocular conjunctiva, it reaches the central nervous system via the trigeminal nerve.” Was this shown? If yes please cite.

Further: “When it comes into contact with the nasal mucosa, SARS-CoV-2 reaches the brain through the olfactory nerve..” This is highly disputed with more data showing that this is not the case.

Further: “… a large route of transfer to the central nervous system is represented by the vagus nerve..”. Again, I have not seen a study clearly showing this.

There are more statements along this line that are not supported by data.

In Figure 3, Spread and mechanisms of CNS damage are described. I do not see the connection to this study and some of the statements are wrong or not backed by data.

The authors write: “SARS-Cov-2 uses transynaptic viral spread (for?) neuroinvasion from periphery to the brain”. Was this shown? If yes please cite.  

Further: “SARS-CoV-2 damages macrophages, microglia and astrocytes.” Is this the case, or are these cells rather activated?

There are more statements that are not supported by data.

Reviewer 2 Report

Very interesting and well written paper on an important topic. However, I have some comments and suggestions which may improve the quality of this manuscript.

  1. Please define long-covid more clearly? Why not post-covid and use the definition of the WHO? Post-COVID-19 is a condition that occurs in individuals with a history of probable or confirmed SARS-CoV-2 infection, usually 3 months from the onset of COVID-19, with symptoms that last for at least 2 months and cannot be explained by an alternative diagnosis
  2. Figure 1: Have you checked for outlyers? Specifically last figure (Interleukin-6).
  3. Table 4: A figure with the correlation would be more helpful.
  4. Add effect sizes (Cohens-d) to the results.
  5. Women are more affected by long-COVID thank men. Any sex differences in your study? If not, I suggest a brief discussion about sex differences (e.g. future studies).

Reviewer 3 Report

The manuscript by Acanfora et al., was a relatively comprehensive research article on the potential roles of blood factors and physiological regulations following the COVID-19 viral infection. The authors seemed to focus on the long-term changes of the patients and summarize the systematic mechanisms. Investigations were well performed, and data was well collected. Due to experimental design issues, the article might be more suitable for more specific journals. There were some moderate concerns:

  • The infection detection was missing/not described in the manuscript.
  • The strain of the SARS-CoV-2 was not reported.
  • The long-term changes were reported but not conclusive. To improve the quality of the research, the patients information should be added regarding any follow-up test.